# Toward More Comprehensive Homologous Recombination Deficiency Assays in Ovarian Cancer Part 2: Medical Perspectives

**DOI:** 10.3390/cancers14041098

**Published:** 2022-02-21

**Authors:** Stanislas Quesada, Michel Fabbro, Jérôme Solassol

**Affiliations:** 1Medical Oncology Department, Institut Régional du Cancer de Montpellier (ICM), 34298 Montpellier, France; michel.fabbro@icm.unicancer.fr; 2Faculty of Medicine, University of Montpellier, 34090 Montpellier, France; j-solassol@chu-montpellier.fr; 3Montpellier Research Cancer Institute (IRCM), Institut National de la Santé et de la Recherche Médicale (INSERM) U1194, University of Montpellier, 34298 Montpellier, France; 4Department of Pathology and Onco-Biology, Centre Hospitalier Universitaire (CHU) Montpellier, 34295 Montpellier, France

**Keywords:** high-grade serous ovarian cancer, homologous recombination deficiency, PARP inhibitors, olaparib, niraparib, rucaparib, companion diagnostic assays

## Abstract

**Simple Summary:**

High-grade serous ovarian cancer (HGSOC—the most frequent and aggressive form of ovarian cancer) represents an important challenge for clinicians. Half of HGSOC cases exhibit homologous recombination deficiency (HRD), mainly through alterations in *BRCA1* and *BRCA2*. This leads to sensitivity to PARP inhibitors, a novel class of breakthrough molecules that improved HGSOC prognoses. To date, three companion diagnostic assays have received FDA approval for the evaluation of HRD status, but their use remains controversial. In this companion review (Part 1: Technical considerations; Part 2: Medical perspectives), we develop an integrative perspective, from translational research to clinical application, that could help physicians and researchers manage HGSOC.

**Abstract:**

High-grade serous ovarian cancer (HGSOC) is the most frequent and aggressive form of ovarian cancer, representing an important challenge for clinicians. Half of HGSOC cases have homologous recombination deficiency (HRD), which has specific causes (mainly alterations in *BRCA1/2*, but also other alterations encompassed by the *BRCAness* concept) and consequences, both at molecular (e.g., genomic instability) and clinical (e.g., sensitivity to PARP inhibitor) levels. Based on its prevalence and clinical impact, HRD status merits investigation. To date, three PARP inhibitors have received FDA/EMA approval. For some approvals, the presence of specific molecular alterations is required. Three companion diagnostic (CDx) assays based on distinct technical and medical considerations have received FDA approval to date. However, their use remains controversial due to their technical and medical limitations. In this companion and integrated review, we take a “bench-to-bedside” perspective on HRD definition and evaluation in the context of HGSOC. Part 1 of the review adopts a molecular perspective regarding technical considerations and the development of CDx. Part 2 focuses on the clinical impact of HRD evaluation, primarily through currently validated CDx and prescription of PARP inhibitors, outlining achievements, limitations and medical perspectives.

## 1. Introduction

Ovarian cancer (which encompasses cancers of the ovaries, fallopian tubes and peritoneum) arises from epithelial cells in 90% of cases, leading to epithelial ovarian cancer; conversely, non-epithelial cancers comprise germ cell tumors, sarcomas, lymphomas and tumors from the granulosa. The term “epithelial ovarian cancer” actually encompasses a heterogeneous group (considering histology and molecular patterns) of cancers, including high-grade serous (HGSOC, which accounts for 70% of cases), endometrioid (10%), clear cell (10%), mucinous (3%) and low-grade serous (<5%) carcinomas [1]. Epithelial ovarian cancers have been divided into type I and II cancers: type I includes endometriosis-related tumors (i.e., endometrioid and clear cell carcinomas), mucinous and low-grade serous carcinomas, while type II includes HGSOCs, undifferentiated carcinomas and carcinosarcomas (these two last classes represent less than 1% of ovarian cancers and are not “epithelial” per se). Interestingly, recent molecular data have challenged this classification system [2]. Each epithelial ovarian cancer subtype has its own features, with distinct epidemiology, genetic risk factors and molecular alterations of carcinogenesis, treatment and prognosis [3,4].

HGSOC is both the most frequent and most lethal subtype. Because of the initial asymptomatic nature of this disease, approximately 75% of patients with epithelial ovarian cancer are initially diagnosed at an advanced stage. This corresponds to stage III–IV cancer, according to the International Federation of Gynecology and Obstetrics (FIGO) staging system. Late-stage diagnosis has a huge impact on prognosis: while the all-stage 5-year overall survival rate is approximately 40%, there is substantial variation between stages I–II and stages III–IV (80–95% versus 10–30% 5-year overall survivals, respectively) [5,6,7]. For late-stage HGSOC, frontline standard treatment mainly includes cytoreductive surgery with taxane-/platinum-based systemic therapy, either through primary debulking surgery or neoadjuvant chemotherapy, which produces an excellent initial response in 70% of patients, a condition defined as platinum sensitivity. Unfortunately, approximately 75% of patients experience recurrence within 3 years, with subsequent platinum resistance, inevitably leading to incurable disease and death [5].

As a consequence, in an effort to improve HGSOC outcomes, targeted therapies have been developed. Bevacizumab, a monoclonal antibody initially developed for metastatic colorectal cancer and which targets vascular epithelial growth factor (VEGF), has been shown to be effective for median progression-free disease (PFS) improvement when used concomitantly with chemotherapy and as maintenance in first-line therapy as well as during HGSOC relapse [8,9,10]. However, bevacizumab does not have a clear effect on overall survival, with only one retrospective analysis suggesting this possibility [11]. In addition to anti-VEGF drugs, a novel class of medication emerged during the past decade, the poly(adenosine diphosphate-ribose) polymerase (PARP) inhibitors, which had a substantial effect on HGSOC prognoses [12,13].

As discussed in the companion paper (Part 1), half of HGSOC cases are characterized by homologous recombination deficiency (HRD), which is one of the predictive markers for PARPi efficiency. HRD can be investigated through dual perspectives: causes and consequence.

In terms of etiology, the main cause of HRD is bi-allelic inactivation of *BRCA1* or *BRCA2* (*BRCA1/2*) genes by either germline (*gBRCA**) plus somatic (*sBRCA**) loss or both functional alleles somatically. Short mutations (single nucleotide or short insertions/deletions) in *BRCA1/2* within the tumor (*tBRCA**), which can reveal either *gBRCA** or *sBRCA** are estimated to be present in approximately 20–25% of cases [14,15]. Furthermore, aberrant methylation of the *BRCA1* promoter (*BRCA1-CpG+*) is reported in approximately 11–15% of HGSOC cases [14,16]. Notably, multimegabase large rearrangements (LRs, also known as structural variants, or SVs) at the *BRCA1/2* loci, leading to HRD, have been reported in approximately 15% of patients [17]. Moreover, HRD can occur in wild-type *BRCA* (*BRCA^wt^*), giving rise to the concept of “BRCAness” (i.e., an HRD phenotype whose cause is not a direct *BRCA* mutation) [18]. Briefly, BRCAness can be caused by biallelic mutations in some homologous recombination-related (HRR) genes, *EMSY* amplification or epigenetic alterations [19,20,21,22,23,24,25]. Apart from *BRCA1/2*, the following list of HRR genes is currently considered as the best characterized to date: *ATM*, *BARD1*, *BRIP*, *CDK1*, *PALB*, *CHEK1*, *CHEK2*, *FANCL*, *PPP2R2A*, *RAD51B*, *RAD51C*, *RAD51D* and *RAD54L* [4,14,19,20].

Schematically, HRD consequences can be classified into three categories: DNA alterations, epigenetic markers and functional phenotypes. DNA mutations are mainly represented by mutational signatures of specific base substitutions and by gross LR and aberrations [26,27,28,29]. Several types of LR have been described as consequences of HRD. Known by the generic term “genomic scars”, they compose a permanent fingerprint of HRD-related global genomic instability [30]. Notably, although “genomic scars” refers to any HRD-related DNA mutation (i.e., encompassing both microlesions and macrolesions), we will restrain this term to large-scale rearrangements as it is used in the literature. Three types of alterations are enriched in tumors with HRD: loss of heterozygosity (LOH), large-scale state transitions (LST) and telomere allelic imbalance (TAI) [31,32,33]. Notably, in the context of HRD, these alterations are found across the genome, and their global enrichment reflects a global genomic instability score (GIS) [34]. In addition to genetics consequences, HRD also leads to specific epigenetic and functional consequences. Clinically, HRD tumors tend to be more sensitive to platinum-based regimens, and specifically to PARPis, through synthetic lethality. Briefly, synthetic lethality relies on the fact that cancer cells harbor gene defects which are not lethal per se, but which turn lethal when combining with a defect in another gene [35]. PARPi leads to SSB accumulation which progresses to DSB. In the context of HRD, cells will accumulate DSB, ultimately leading to apoptosis [13,36]. HRD HGSOC cases tend to have improved progression-free and overall survivals, which is in part due to better responses to treatment [37]. Consequently, an accurate evaluation of patients’ HRD status is essential for both prognosis and therapeutic choice.

To meet this demand, several companion diagnostic (CDx) assays have been developed and clinically validated, leading to substantial improvements in the management of HGSOC. However, their use remains controversial, notably because they do not assess directly HRD status but their causes and/or their consequences. While Part 1 of this companion review focuses on the molecular and technical considerations of HRD definitions and assessments, Part 2 aims to bridge the gap between technical and clinical perspectives. Thus, we will review: 1. the added value of CDx integration, to date, according to published clinical trials that assessed PARPis, with a bioclinical perspective of molecular status, clinical situation and subsequent approvals; 2. the inherent limits of current CDx assays, notably regarding medical considerations; and 3. proposed perspectives regarding these drawbacks to improve management of HGSOC.

Combined with the companion paper (Part 1), this state-of-the-art and current perspectives review will provide clinicians and researchers with a translational and integrated view of HRD in HGSOC.

## 2. HRD Companion Assays in Clinical Practice

### 2.1. Introduction to CDx Assays

Owing to the potent impact of HRD on HGSOC management, several CDx assays have been developed and validated. While the companion paper focuses on an extensive description of technical considerations (performances and limitations of each test), this paper will detail the clinical side (i.e., the relevance of evaluating HRD status as a biomarker for PARPi prescription and expected treatment response). To date, 3 CDx assays are currently FDA-approved for epithelial ovarian cancer cases: BRACAnalysis^®^ CDx (BA-CDx), developed and marketed by MyriadGenetics (MG, Salt Lake City, UT, USA), FoundationFocus CDxBRCA-LOH^®^ (FF-CDx) from FoundationMedicine (Cambrige, MA, USA) and MyChoice CDx^®^ (MC-CDx) from MG [38]. As a reminder, HRD evaluation mainly relies on two strategies: searching for mutations in HR-related genes (mainly *BRCA**; the causes of HRD) and/or the presence of “genomic scars” (i.e., the consequences of HRD).

BA-CDx relies on *gBRCA* assessment through sequencing genomic DNA obtained from whole blood samples collected in EDTA [39]. BA-CDx is currently FDA-approved for distinct situations (discussed below) where detection of *gBRCA** is required prior to PARPi prescription.

In addition to BA-CDx, FF-CDx and MC-CDx assays assess both *tBRCA* and genomic scars within the tumor [40,41]. Notably, as *BRCA* analysis is performed on tumoral tissue, it does not distinguish between *gBRCA** and *tBRCA**. Genomic scars are evaluated very differently between FF-CDx and MC-CDx.

For MC-CDx, HRD status is based on a proprietary “genomic instability score” (GIS), consisting of the unweighted numeric sum of LOH, LST and TAI. HRD positivity is currently defined by a GIS score ≥ 42 and/or *tBRCA1/2** [42].

For FF-CDx, HRD status is based on genome-wide evaluation of LOH, leading to a global score reflecting the percentage of genomic LOH: tumors are defined as “LOH high” (≥16%) or “LOH low” (<16%), corresponding to HRD positive (HRD+) and negative (HRD−) statuses, respectively [43]. Notably, a positive HRD result is indicated if the tumor is “LOH high” and/or exhibits a *tBRCA**. Importantly, FF-CDx is no longer available as a stand-alone assay but is included within the more general FoundationOne CDx© (F1-CDx), a comprehensive multicancer analysis with 324-gene panel testing coupled with analysis of tumor mutational burden, microsatellite instability and specific gene rearrangements [41]. In addition, FoundationMedicine recently marketed the FoundationOne Liquid^®^ CDx, which detects *tBRCA** directly in a blood sample through 324-gene panel testing; this CDx does not evaluate LOH [44]. Although FF-CDx (for *tBRCA** detection) is required for PARPi prescription in two distinct indications in OC, the third FDA-approved indication (i.e., HRD evaluation prior to rucaparib maintenance in the second-line setting) is not biomarker driven, as HRD positivity is predictive of efficacy and indicates enhanced PFS.

Owing to their clinical impact, FF-CDx and MC-CDx (both FDA-approved assays) have been included as part of epithelial ovarian cancer management in the recommendations of the European Society for Medical Oncology (ESMO) and the American Society of Clinical Oncology (ASCO) [40,45]. However, a recently published expert consensus identified the urgent need to develop new tools that can more accurately assess HRD status in HGSOC cases [46].

### 2.2. Assay Achievements in Clinical Practice: The PARPi Era

#### 2.2.1. Introduction to Clinical Trials Assessing PARPis

The past decade led to a breakthrough in the management of OC, mainly owing to the introduction of PARPis. Initial proof-of-concept studies showed that PARP inhibition leads to synthetic lethality in *BRCA*-deficient cells [47,48]. Subsequently, the first clinical studies were performed in patients harboring *gBRCA** with multitreated tumors (including OC). Patients who were treated with olaparib had objective antitumor activity; specifically, this antitumor activity in *gBRCA** OC was associated with platinum sensitivity [49]. These encouraging results paved the way for many RCTs on OC and consequently FDA/EMA approvals in distinct clinical contexts and molecular situations (Table 1). Indeed, there are currently three FDA/EMA-approved PARPis for OC management: olaparib (Lynparza^®^, AstraZeneca, Cambridge, UK), niraparib (Zeluja^®^, GlaxoSmithKline, Brentford, UK) and rucaparib (Rubraca^®^, Clovis, Boulder, CO, USA).

This section will describe the main results of RCTs that led to PARPi approval, with a focus on the analyses performed (and the *BRCA* and HRD statuses) when applicable. Importantly, several subgroup analyses of the studies described here were performed as predefined exploratory endpoints; thus, their results should not be taken as absolute. An extensive description (e.g., the clinical characteristics of the enrolled patients) of these clinical studies is beyond the scope of this review and can be found elsewhere. Caution should be used when considering comparisons between these clinical studies, as their eligibility criteria and molecular evaluations differed.

Unless stated otherwise, all FDA/EMA approvals are for advanced/recurrent epithelial ovarian cancer (including primitive peritoneal and fallopian tube cancers); furthermore, the EMA restricts these treatments to high-grade cancers. Notably, frontline and second-line maintenance therapies are based on selected patients who had complete/partial response (C/PR) to platinum-based chemotherapy. These responses are defined through radiology and biology parameters: complete response indicates the disappearance of all measurable/assessable pathologies that were present at the start of chemotherapy and normalization of CA-125 levels, while clinical partial response indicates situations where radiologic evidence of disease and/or an abnormal CA-125 level remained after treatment.

Clinical studies and market approval started with multitreated OC and progressively reached frontline management. For clarity, this section will be divided into three distinct subsections: frontline management, first recurrence and second recurrence and beyond.

#### 2.2.2. Newly Diagnosed Advanced Epithelial Ovarian Cancer: Frontline Treatment

There are several possibilities for first-line treatment given as maintenance monotherapy (1 Lm), depending on the molecular context. Olaparib was evaluated through the SOLO1 (NCT01844986) RCT, which included 391 patients with HGSOC (or nonserous carcinomas in 5% of cases) or high-grade endometrioid carcinomas with *tBRCA**, either tested locally (n = 210) or prospectively with BA-CDx (n = 178) or through BGI clinical laboratories (n = 3); notably, 388 patients had *gBRCA** [50]. Olaparib had a 70% lower overall risk of cancer progression or death (hazard ratio (HR) = 0.30; 95% confidence interval (CI): 0.23–0.41; *p* < 0.001) compared to the placebo (in a 2:1 ratio; 260 patients received olaparib, and 131 received placebo) once patients had at least a partial response after carboplatin-paclitaxel chemotherapy. This seminal work led to the first FDA and EMA approvals (in 2018 and 2019, respectively) for PARPis as a 1 Lm monotherapy in patients with *gBRCA** or *tBRCA** (tested with the FDA-approved BA-CDx and F1-CDx arrays, respectively). Recently, the 5-year follow-up of SOLO1 showed a median PFS of 56 versus 13.8 months (HR = 0.33; 95% CI: 0.25–0.43) with olaparib and the placebo, respectively [51].

The PAOLA-1 (NCT02477644) RCT compared bevacizumab plus olaparib (OlaBeva) versus bevacizumab plus placebo as a 1 Lm [52]. A total of 806 patients with HGSOC (or nonserous carcinomas in 5% of cases) were enrolled and prospectively tested with the MyChoice HRD assay (analogous to the MC-CDx but dedicated to the research field), which allowed a comparison of PFS according to HRD status, after surgery and carboplatin-based chemotherapy. Interestingly, 30% of enrolled patients had a *tBRCA* deleterious mutation, and 50% were HRD+ (with a GIS ≤ 42 cutoff score for positivity). OlaBeva led to a median PFS of 22.9 months (versus 16.6 months with bevacizumab alone; HR = 0.59 95% CI: 0.49–0.72; *p* < 0.001). Interestingly, subgroup analyses showed a greater effect of the addition of olaparib in specific populations. In patients with *tBRCA**, the median PFS was 37.2 months with OlaBeva versus 21.7 months with bevacizumab alone (HR = 0.31; 95% CI: 0.20–0.47). In the HRD+ patients, the median PFS was 37.2 months with OlaBeva (versus 17.7 months with bevacizumab alone; HR = 0.33; 95% CI: 0.25–0.45). This positive effect persisted even in HRD+ patients that lacked a *BRCA* mutation (28.1 months with OlaBeva versus 16.6 months with bevacizumab alone; HR = 0.43; 95% CI: 0.28–0.66), suggesting that patients with *tBRCA** saw the greatest benefit. Anecdotally, the positive effect on PFS was slight in patients with an unknown HRD status but nonexistent in the HRD− patients, suggesting that only the latter did not benefit from treatment with OlaBeva. This led to FDA/EMA approval for olaparib plus bevacizumab in 2020 as a 1 Lm therapy in the HRD+ population, with concomitant FDA approval for the MC-CDx assay. Interestingly, the SOLO1 and PAOLA-1 studies raised the question of whether olaparib alone had an effect in HRD+ patients (irrespective of *tBRCA** status), as it was not directly tested.

The PRIMA study (NCT02655016) enrolled 733 patients with macroscopic residual disease and investigated their sensitivity to platinum-based chemotherapy as well as comparing niraparib and a placebo as a 1 Lm monotherapy, both in the context of HRD+ and in the overall population [50]. HRD status was prospectively tested with MC-CDx: 373 (50.9%) patients had HRD+ tumors, of whom 223 (59.7%) had *tBRCA**. In the HRD+ group, the median PFS was 21.9 months with niraparib versus 10.4 months with placebo (HR = 0.43; 95% CI: 0.31–0.59; *p* < 0.001). Interestingly, the impact of *BRCA* status was slight in this study, as the PFS was 22.9 months (versus 10.9 months with placebo; HR = 0.40) in patients with *tBRCA** and 19.6 months (versus 8.2 months with placebo; HR = 0.50) in patients with *BRCA^wt^*. In patients with HRD negative tumors, the PFS was still slightly higher at 8.1 months (versus 5.4 months with placebo; HR = 0.68). Finally, in the entire sample, niraparib increased the median PFS (13.8 months with niraparib versus 8.2 months with placebo; HR = 0.62; 95% CI: 0.50–0.76; *p* < 0.001). As such, this latter point led to FDA/EMA approvals of 1 Lm niraparib irrespective of biomarker status.

In addition, although it is not FDA/EMA approved yet, veliparib has been tested through the VELIA (NCT02470585) RCT as 1 L throughout (chemotherapy plus veliparib followed by veliparib maintenance) versus chemotherapy, with prospective evaluation of HRD status with MC-CDx [53]. In VELIA, the GIS cutoff was lowered to a 33 ≥ GIS cutoff score for positivity to include patients with a putative HRD+ status. The primary endpoint was median PFS: the *tBRCA** group exhibited the greatest benefits of veliparib throughout versus chemotherapy (34.7 versus 22.0 months; HR = 0.44; 95% CI: 0.28–0.68; *p* < 0.001), compared to the HRD+ group (31.9 versus 20.5 months; HR = 0.57; 95% CI: 0.43–0.76; *p* < 0.001) and the intention-to-treat (ITT) group (23.5 versus 17.3 months; HR = 0.68; 95% CI 0.56–0.83; *p* < 0.001). Although an exploratory analysis was performed, the HRD− group did not seem to benefit from veliparib (15.0 versus 11.5 months; HR = 0.81; 95% CI: 0.60–1.09).

Overall, PARPis as 1 Lm treatment has become the standard according to American and European clinical guidelines: olaparib is recommended in the context of *tBRCA** and olaparib plus bevacizumab is recommended in HRD+ tumors. Furthermore, in HRD-negative patients (i.e., with proficient HR), niraparib should be considered a maintenance therapy option [45,54,55].

#### 2.2.3. Recurrent Epithelial Ovarian Cancer: Second-Line Maintenance

For second-line treatment and beyond, platinum sensitivity is measured by the platinum-free interval, which is the interval from the date of last platinum dose until progressive disease is documented; platinum sensitivity guides second-line treatment with platinum or PARPis. Depending on the duration of remission, recurrence can be “rechallenged” with platinum or PARPis. A platinum-free interval shorter than 6 months is defined as platinum resistance, but a platinum-free interval longer than 6 months refers to a more heterogeneous group, including “true” (i.e., ≥12 months) and partial (i.e., within 6–12 months) sensitivity [56].

In the first relapse after initial treatment, recurrent epithelial ovarian cancer has simpler FDA/EMA approvals for PARPi [57]. Indeed, niraparib, olaparib and rucaparib can be used as second-line maintenance monotherapies (2 Lm) in patients with recurrent epithelial ovarian cancer in complete to partial response to platinum-based chemotherapy, independent of molecular status (i.e., *gBRCA**, *tBRCA** or HRD+).

The ARIEL3 (NCT01968213) RCT (n = 564) evaluated rucaparib versus placebo as a 2 Lm in a prospective molecularly-defined cohort according to FoundationMedicine T5-NGS testing; furthermore, *gBRCA* status was evaluated through BA-CDx [58]. HRD was considered positive if LOH was ≥16 and/or in the presence of *tBRCA**. A total of 564 patients were enrolled, with the following molecular characteristics: 196 with *tBRCA** (of whom 130 had *gBRCA**), 158 HRD+ without *tBRCA**, 161 HRD-negative and 49 undefined. Rucaparib was evaluated through PFS with the use of an ordered step-down procedure for three nested cohorts. Enhanced PFS was described in the three cohorts as follows: *tBRCA** (16.6 months versus 5.4 months for placebo, HR = 0.23; 95% CI: 0.16–0.34; *p* < 0.0001), HRD+ (13.6 months versus 5.4 months for placebo, HR = 0.32; 95% CI: 0.24–0.42; *p* < 0.0001) and ITT (10.8 months, versus 5.4 months for placebo, HR = 0.32; 95% CI: 0.30–0.45; *p* < 0.0001). Although they were not directly compared, rucaparib seemed to exert a greater effect in the context of HRD+ or *tBRCA**. Interestingly, the authors reported that more than 30% of HRD− patients still benefited more than a year after rucaparib treatment. Given the ARIEL3 results, rucaparib received FDA/EMA approval as 2 Lm regardless of patient molecular status; F1-CDx later received FDA approval for LOH evaluation, allowing clinicians to determine which patients would benefit the most from this maintenance therapy [59].

The NOVA (NCT01847274) RCT evaluated niraparib as a 2 Lm in two independent cohorts based on *gBRCA* status, as determined by BA-CDx [60]. This resulted in the enrollment of 553 patients: 203 with *gBRCA** and 350 without *gBRCA**. The latter group consisted of 162 HRD+ patients, of whom 47 (29.0%) had *tBRCA**, 134 HRD− and 54 were undetermined for HRD status. Niraparib led to significantly longer PFS in the two distinct cohorts: 21.0 versus 5.5 months in the *gBRCA** cohort (HR = 0.27; 95% CI 0.17–0.41; *p* < 0.001) and 12.9 versus 3.8 months (HR = 0.38; 95% CI, 0.24–0.59; *p* < 0.001) in the HRD+ non-*gBRCA** cohort. Subsequently, the global non-*gBRCA** cohort showed enhanced PFS with 9.3 versus 3.9 months (HR, 0.45; 95% CI: 0.34–0.61; *p* < 0.001). Finally, exploratory analyses showed that in the predefined subgroups (HRD+ with *sBRCA**, HRD+ without *sBRCA** and HRD−), niraparib still led to enhanced PFS. Therefore, the NOVA RCT led to niraparib approval as a 2 Lm for epithelial ovarian cancer.

The SOLO2 (NCT01874353) RCT included 295 recurrent epithelial ovarian cancer cases with *gBRCA** (evaluated through BA-CDx) and showed an improved PFS with olaparib of 19.1 versus 5.5 months (HR = 0.30; 95% CI: 0.22–0.41; *p* < 0.0001) [61]. Furthermore, Study 19 (NCT00753545) randomized 265 patients regardless of *BRCA* mutation and demonstrated an improvement in PFS in patients treated with olaparib versus placebo (HR = 0.35; 95% CI: 0.25–0.49; *p* < 0.0001). These two RCTs led to approval of olaparib as a 2 Lm independent of molecular status [57]. Notably, according to ARIEL3 and NOVA data, HRD status was not predictive of PARPi benefit; indeed, even HRD- patients still had an improved PFS.

#### 2.2.4. Recurrent Epithelial Ovarian Cancer: Third Line and Beyond

In contrast to their use in 1 Lm and 2 Lm, PARPis in the third line and beyond are used as monotherapies (i.e., not subsequent to platinum-containing chemotherapy). Within the context of recurrent epithelial ovarian cancers already treated with ≥2 lines of chemotherapy, three PARPis are currently approved but have distinct molecular requirements.

In 2014, olaparib was the first PARPi to ever receive FDA approval for recurrent OC with *gBRCA**; patients needed to be previously treated with ≥3 lines of chemotherapy regardless of platinum sensitivity. This approval was based on Study 42 (NCT01078662), a single-arm international trial that enrolled 137 patients previously treated with ≥3 lines of chemotherapy that carried *gBRCA**; patients showed an objective response rate of 34% and a median response duration of 7.9 months with olaparib. As described in the previous section, *BRCA* status was initially tested locally to assess patient eligibility and retrospectively evaluated with BA-CDx in a clinical bridging study, leading to its approval as a CDx in this context [39,62].

Rucaparib FDA approval was based on two open-label, single-arm trials, Study 10 (NCT01482715) (n = 42) and ARIEL2 (NCT01891344) (n = 64) [63,64]. Study 10 was a phase I/II study that evaluated the safety and efficacy of rucaparib (part II recruited OC patients with *gBRCA** or *tBRCA** who were previously treated with 2–4 prior lines) through objective response rate and evaluation of median response duration. ARIEL2 was a phase II study that evaluated rucaparib efficacy in OC patients carrying *tBRCA** (Part I enrolled 204 patients with ≥1 L and platinum sensitivity, while Part II recruited 111 heavily pretreated patients). Interestingly, patients from ARIEL2 were molecularly categorized into three groups through FoundationMedicine T5 NGS: HRD+ with *tBRCA**, HRD+ without *tBRCA** (defined in this study as LOH ≥ 14% or “LOH high”) and HRD− (LOH < 14% or “LOH low”). Patients from Part I included the following: 40 HRD+ patients with *tBRCA**, 82 HRD+ patients without *tBRCA**, 70 LOH-low patients and 12 undefined patients. In addition to assessing the efficacy of rucaparib, another aim of the study was to evaluate tumor LOH as a predictive biomarker for rucaparib efficacy. Compared to the LOH-low subgroup (which exhibited a PFS of 5.2 months), PFS was significantly longer in the *tBRCA** (PFS of 12.8 months, HR = 0.27; 95% CI 0.16–0.44, *p* < 0.0001) and LOH (PFS of 5.7 months, HR = 0.62; 95% CI: 0.42–0.90, *p* < 0.01) subgroups, suggesting the efficacy of rucaparib for patients with *tBRCA**.

Given the superiority of *tBRCA** over HRD as a predictor of longer PFS and with the aim of testing broader applications, an amendment to the study protocol was made, leading to the inclusion of heavily pretreated OC patients in Part II of the study. Subsequently, FDA approval of rucaparib was based on a review of efficacy in 106 patients (combining Study 10 and both ARIEL2 cohorts and including patients with recurrent epithelial ovarian cancers carrying *tBRCA** who received ≥ 2 prior platinum-containing lines, including patients with platinum-resistant or refractory status), that found an objective response rate of 54% and a duration of response of 9.2 months. Consequently, analysis of ARIEL2/Study-10 led to the approval of rucaparib in recurrent epithelial ovarian cancers patients with *tBRCA** previously treated with ≥3 lines of chemotherapy and regardless of platinum sensitivity. The retrospective analysis of *tBRCA** by FF-CDx as a clinical bridging study with 67 of patients (and a 96% concordance rate with local testing) led to FDA approval of FF-CDx for identification of patients eligible for rucaparib treatment due to *tBRCA** detection.

More recently, the QUADRA (NCT02354586) single-arm phase II trial tested niraparib as ≥fourth line through the inclusion of HGSOC patients pretreated with ≥3 lines, with different statuses regarding platinum (sensitive, resistant or refractory, defined by the platinum-free interval from last platinum-containing chemotherapy) and tested with MC-CDx [65]. The patients included 63 HRD+ patients with *tBRCA**, 126 GIS ≥ 42 patients without *tBRCA**, 186 HRD− patients and 44 HRD-undefined patients. Niraparib efficacy was evaluated according to objective response rate. In the *tBRCA** subgroup, patients with platinum sensitivity had a 39% objective response rate, while platinum-resistant and platinum-refractory patients achieved 29% and 19%, respectively. In contrast, 20% of HRD+ patients without *tBRCA** with platinum sensitivity achieved a response; this rate dropped to 10% in platinum-resistant/-refractory patients. Of note, HRD-negative/unknown patients exhibited a 3% objective response rate. Consequently, the FDA approved niraparib as ≥fourth-line monotherapy in patients either carrying *tBRCA** or that had GIS ≥ 42 and potentially platinum sensitivity (platinum sensitivity was defined in this study as platinum-free interval ≥ 6 months).

In conclusion, while *gBRCA*/*tBRCA* evaluation appears to be a clear prerequisite for ≥third-line monotherapy with PARPis, the QUADRA study challenges the relevance of HRD status compared to *tBRCA** status, as platinum sensitivity appeared more important.

## 3. HRD Evaluation in Clinics: Current Limitations

### 3.1. General Considerations

Despite producing considerable advances in HGSOC management, HRD validated assays currently suffer from several limitations that fall into two categories: technical or medical concerns (Table 2 and Table 3, respectively).

In general, it should be noted that in the US, only MC-CDx and F1-CDx are FDA-approved, restricting this test to private companies in the US market in terms of clinical practice, with a reduction in shared data, especially regarding technical considerations and classification of variants. Furthermore, these CDx do not have negligible costs, ranging from USD 4040 to USD 5800, and the estimated delay between testing and results is approximately 2 weeks. In Europe, EMA approvals do not depend on specific CDx for HRD identification; while private companies (such as MG) exist, institutional laboratories are currently developing their own assays based on GIS, with the aim of providing better patient access, affordability and inter-institutional exchanges. Specifically, an international project driven by the European Network of Gynecological Oncological Trial (ENGOT) is under development. The cost of BA-CDx is low in the US, as genetic testing services for hereditary cancer are covered by most health insurance companies; furthermore, MG provides a financial assistance program.

Beyond these general aspects, clinical considerations of HRD assays occur on at least three distinct levels: the tissue (i.e., the analyzed sample), the result (i.e., the relevance of HRD as a biomarker) and the clinical context (i.e., the consideration of medical and evolutionary dimensions).

### 3.2. A Matter of Tissues

As described in the preanalytical section, tumor samples frequently exhibit distinct subclones within tumors, and any given result will be a synthesis of these subclones according to their relative proportions [66,67,68]. By definition, a biopsy represents only a fraction of the tissue at the site. Thus, a sample could be HRD+ while actually consisting of a mix of HRD+ and HRD− components. This latter component may participate in primitive resistance (or at least reduced sensitivity) to PARPis/platinum salts and/or to recurrence through mechanisms of clonal selection and expansion. Similarly, a discrepancy between primitive tumors and their metastases or even between distinct metastases indicates the complexity of evaluating HRD status and subsequent HGSOC management.

Apart from the HRD evaluation itself, the epigenomic and genomic contexts in an HRD+ sample should be considered. As explained in the first section, many layers of complexity can promote a deficient or proficient HR status. Furthermore, HRD+ tumors are not a homogenous group and require classification that is more precise. While *gBRCA** or *tBRCA** with GIS ≥ 42% in a newly diagnosed HGSOC produces high confidence of the true HRD+ status, the (epi)genomic context also modulates the functional HRD status and should not be overlooked [69]. In parallel with the polygenic risk score that modulates OC risk in patients with *gBRCA**, a similar score could potentially modulate HRD phenotype [70].

In addition to the tumor itself, the cellular microenvironment should be taken into consideration, although its complexity is clearly beyond the scope of this review. Data suggest that PARPis can exhibit different activity depending on the tumor microenvironment, such as a hypoxic condition, a concept called “contextual synthetic lethality” [71]. Furthermore, a complex interaction exists between microenvironment status and tumor-infiltrating immune cells (and notably T cell receptor clonality) that could modulate the functional effects of HRD [72].

### 3.3. Relevance of HRD Status as a Predictive Biomarker

In addition to technical considerations of false-positive/negative cases (FP/FN), HRD assays, although predictive at the population scale, still suffer from false-positive/negative cases during clinical applications. In other words, some patients will respond to PARPis although they are HRD−, while HRD+ patients will not. Therefore, HRD should not be considered a “black box” with a simplified dichotomy (i.e., proficient versus deficient). For instance, RCTs that evaluated PARPis globally showed a higher effect in the context of *BRCA1/2** than in the context of *BRCAness*. Relationships between a given gene harboring a pathogenic variant and patient sensitivity to therapeutics should be analyzed further in the future. For instance, in OC patients with *gBRCA2**, only cases with mutations affecting the RAD51 binding domain exhibited improved overall survival [73]. Furthermore, a study focused on long responders under olaparib maintenance treatment showed that an enrichment of *BRCA2** affected the RAD51-binding domain within this group [74].

Notably, a vast majority of RCTs assessing PARPis as first and second lines that subsequently led to FDA/EMA approvals included patients who were platinum sensitive (evaluated through complete/partial response to platinum-based chemotherapy regimens), thus selecting specific populations that already exhibited a sensitivity profile close to PARPis [75,76].

In terms of frontline treatment, HRD (through MC-CDx) evaluation in PAOLA-1 patients appeared essential to guide treatment with olaparib plus bevacizumab, but niraparib exerted an effect irrespective of HRD status [52,76,77]. Interestingly, patients from the PRIMA trial were quite different from those from the PAOLA-1 trial, with more advanced cancer and less complete response. Nevertheless, niraparib efficiency could be explained in part by an “HRD-unrelated” effect [78].

Moreover, second-line RCTs for recurrent epithelial ovarian cancers failed to clearly account for CDx, as all PARPis are “biomarker-agnostic” approved. When niraparib was evaluated in the fourth-line setting within the platinum-resistant population (according to the platinum-free interval), *BRCA1/2** patients still exhibited benefits, but non-*BRCA** HRD+ patients showed almost no effect. This emphasizes the importance of platinum status in the latter condition, as it clearly outperformed HRD status as a biomarker for PARPi sensitivity [65]. Conversely, regarding rucaparib and olaparib approvals in the third-line settings, the presence of *BRCA** (germinal and tumoral, respectively) is the only prerequisite for their prescription, irrespective of patients’ platinum status. These different results emphasize the importance of the timing of GIS evaluation and its variable predictive value.

A GIS threshold of ≥42 also raises questions as to its clinical relevance. Indeed, a binary cutoff increases the likelihood of missing patients who could benefit from PARPis and selecting patients that would not benefit from PARPis, especially among those with near-threshold values. Furthermore, an identical threshold value has been defined for both frontline and ≥third-line treatment, but through evolution, HRD tumors will likely increase their GIS by accumulating an increasing number of genomic scars.

### 3.4. An Evolutionary Perspective on HRD Status

The main drawback, as previously explained, is that GIS/LOH scores rely on genomic scars accumulated over time, which are the consequences of past HRD, irrespective of current HRD status. In addition to the classic limitations of a biomarker, such as sensitivity, specificity and FPs/FNs, HRD evaluated via genomic alterations needs to take into account the evolution of HGSOC.

Analysis of initial samples before frontline treatment is not shaped by chemotherapy (which kills sensitive clones and potentially allows the emergence of resistant clones) and classically reveals the “native” state of the tumor; thus, evaluating HRD after several previous lines of chemotherapy leads to possibly irrelevant results [79,80,81,82,83,84]. Indeed, the sensitivity/resistance switch of cancerous cells is a dynamic phenomenon during clonal expansion from pre-existing clones or through new (epi)mutations. As such, GIS measures the initial state of the tumor, naïve to treatments; it can be either concordant or discordant with the tumor’s current status. The further a patient is from frontline treatment, the higher the risk of discordance between GIS and actual status, as therapies mainly rely on platinum. Indeed, as platinum salts generate DSBs, they will treat clones with HRD but HR proficient clones will be spared. This discordance has been highlighted in the QUADRA study.

Discordance between HRD status (i.e., *tBRCA** and/or high LOH/GIS) and PARPi efficiency can be the consequence of several mechanisms. The most characterized mechanism is reverse mutation within *BRCA1/2* genes. Indeed, it has been shown that during HGSOC treatment with PARPis, reverse mutations appeared within *BRCA1/2*, leading to restoration of functional proteins and thus an HR-proficient profile, an event correlated with drug resistance [85]. Interestingly, reverse mutations within *BRCA1/2* are not universal. Indeed, a differential “reversion potency” exists between distinct mutation states. Hypermethylation of *BRCA1* promoter, which relies on a dynamic process, is an easily removed epigenetic mark; interestingly, the context of heterozygous methylation is associated with resistance compared with that of homozygous hypermethylation. On the other hand, LRs or biallelic losses of *BRCA1/2* are considered almost irreversible and could explain, at least in part, the prolonged response to PARPis in some patients [74,86]. Point mutations within *BRCA1/2* leading to coding sequence disruption depend on the mutation rate in cancer cells. Interestingly, NHEJ should be viewed as having dual properties: while it is essential for the synthetic lethality of PARPis during DSB generation, it also exerts a promutagenic effect that favors the emergence of resistant clones [87]. Strikingly, *BRCA* reverse mutations present at pretreatment circulating tumoral DNA have been shown to predict resistance to rucaparib [88].

As such, *BRCA** should not be considered wild-type versus mutated but rather include mechanistic considerations. Although *gBRCA** and *tBRCA** carriers appear to exhibit the same benefits from PARPi treatment on a population scale, this requires deeper analysis [89]. Indeed, even the *gBRCA** pathogenic variant, the strongest biomarker for PARPi sensitivity thus far, should not be considered a definitive biomarker; studies assessing olaparib efficiency in *gBRCA** carriers showed that the objective response rate was seen in approximately 40% of patients, underscoring the fact that sensitivity to PARPis is a complex phenomenon, with intermingled HRD-related and unrelated pathways [90,91].

In addition to *BRCA*-related HRD, several other mechanisms have been shown to counteract sensitivity to PARPis, such as secondary restoration of *RAD51C/D* [92]. Although it is far beyond the scope of this review, we should note that fundamental research on cell lines or mouse models has described several genetic/epigenetic mechanisms (e.g., PARP1 mutations, miRNA-622 overexpression) that explain PARPi resistance, mainly based on HR restoration, DNA replication fork protection, drug efflux through overexpression of multidrug resistance protein (MDR1) and impact of the tumor microenvironment [93,94,95,96]. Evidence-based *in cellulo* research has even suggested that the HR process should be viewed on a continuum, including influences of (epi)genetic modulation (through amplification and/or overexpression) between pathway choices concerning DNA repair [97].

To further complicate the analysis, discordance between sensitivity to platinum salts and PARPis has been shown. This can be the consequence of acquired resistance through platinum treatment. However, a specific subset of patients with platinum sensitivity but no primitive resistance to PARPis, due to mutations in the nucleotide excision repair pathway, has been described [98]. Although described *in cellulo*, defects in the NHEJ pathway lead to rucaparib resistance [99]. In contrast, some patients, notably those carrying *CDK-12* mutations, exhibit a low LOH, contrasting with their HRD profile [100].

In summary, several dimensions should be taken into account when studying these phenomena: primary/acquired resistance, *BRCA*-related/unrelated mutations, the type of *BRCA* alterations, and HRD-related/unrelated resistance mechanisms.

## 4. Emerging Strategies and Perspectives for Accurate and Dynamic Assessment of HRD

### 4.1. Introduction

As described in Part 1 of this companion review, emerging strategies mainly focus on three axes: other molecular tools for HRD assessment (i.e., apart from *BRCA1/2* and LOH/GIS scores), dynamic assays (i.e., functional assays) for evaluating current HRD status and more global strategies (including nomograms). At the patient scale, several predictive nomograms have been developed, aiming to assist clinicians to estimate the prognosis of HGSOC patients in specific situations or with specific platinum sensitivities [101,102,103,104,105]. Interestingly, a nomogram of the predicted PFS upon the addition of maintenance olaparib therapy in patients with recurrent epithelial ovarian cancers with *BRCA*-mutations and platinum sensitivity has recently been validated [106]. Other nomograms with biological (including HRD), pathological and clinical data should be developed and validated for a more specific prescription of PARPis.

From a more integrated perspective, histopathological markers such as tumor infiltrating lymphocytes (TILs) have been shown to be correlated with clinical outcome [107,108,109]. More precisely, higher levels of CD8+ T cell infiltration have been described in the context of *gBRCA1/2**, and HGSOC patients with HRD have higher CD3+ TILs [110]. Although they are still in the early stages and restricted to research, radiomics (notably linked with biology through proteomics/radiomics correlations) and artificial intelligence-driven projects could help clinicians improve HGSOC management in the near future [111,112].

The proposed strategies and perspectives according to clinical context (i.e., frontline versus second line and beyond) are summarized in Table 4.

### 4.2. Frontline Maintenance Perspectives

From a personalized medicine perspective, HRD evaluation should take into account several parameters. MC-CDx represents the current and FDA-approved cornerstone of HRD evaluation. The first question is which analysis we should perform upon the diagnosis of HGSOC, that is, the priority of *gBRCA* or *tBRCA*. Indeed, while current international guidelines recommend *gBRCA* testing for all patients diagnosed with HGSOC, this point raises several questions, such as access to genetic counseling, costs and delay prior to achieving results [113]. Furthermore, a few cases of *gBRCA** status, but the lack of its presence within tumors, have been described. On the other hand, direct *tBRCA** assessment through MC-CDx allows a quick answer to eligibility for PARPis, with the possibility to secondarily orient the patient to genetic counseling in case of *tBRCA**. Thus, the question raised is: do we need to directly assess GIS (one-step strategy), or should it be performed in case of *tBRCA^wt^* (two-step strategy)?

In addition to establishing this strategy, several refinements should be considered. First, current CDx should modulate their pipelines and integrate a deeper analysis of LRs, which have recently been shown to be an important part of *BRCA1/2* alterations, leading to false negatives if tests are focused on tumor samples [17,114]. Second, HRD evaluation should consider the type of *BRCA* alterations found and their corresponding predictive values. Indeed, *BRCA** are considered based on their (likely) pathogenic status. However, as shown with *BRCA2*, not all mutations are equal [73]. Other signatures, such as miRNAs, gene expression profiles and proteomics, although they have promising initial results, still suffer from “bench-to-bedside” applications and a lack of “confidence” from the clinical side, as they rely on molecules that are dynamic and not clearly defined, in contrast with genetic mutations. As discussed above, a refinement of HRD could aid clinicians when determining whether to prescribe PARPis: for instance, what should clinicians do when faced with a GIS of 40 or 43? In general, the “one fits all” cutoff should be reconsidered, as distinct pathological processes probably lead to distinct molecular features. Although several GIS cutoffs have been proposed, such as ≥33 and ≥62, it is currently too early to rely on definite values; more results could be reached through use as a grayscale.

For the 10–20% of HRD-inconclusive cases, supplementary biomarkers should be considered; for instance, *EMSY* amplification is one of the HRD-causing alterations. Conversely, *CCNE1* amplification is a specific alteration that is almost exclusive to HRD. As such, this could be a useful biomarker with negative predictive value for sensitivity to PARPis [115,116]. Similarly, indirect assessment with molecular signatures (genetic or epigenetic signatures) associated with platinum resistance could help classify patients with inconclusive or conflicting results [117].

Interestingly, in addition to predicting sensitivity to PARPis, HRD evaluation could also be useful for bevacizumab; indeed, while there is no benefit in overall survival for patients who received bevacizumab compared with that of receiving chemotherapy alone, *BRCA** and HRD patients appear to benefit [11]. Functional assays and examination of primary RAD51 foci have promising results, but their implementation would require standardization and development within pathology platforms [118]. Ideally, nomograms similar to existing nomograms on recurrent epithelial ovarian cancer and olaparib could be constructed; these would be a “clinician-friendly” method for considering the construction of a comprehensive nomogram that would integrate clinical (e.g., complete versus partial response to platinum, CA125) and biological (e.g., *gBRCA*/*tBRCA* status, type of mutation, presence of reverse mutations, GIS/LOH, functional assays, comprehensive–omics) data, finding an equilibrium between performance and routine clinical feasibility.

Finally, patients receiving PARPis should receive dynamic assessments, similar to those for other malignancies that are monitored through circulating tumoral DNA. For instance, side effects of PARPis have been described as potential surrogate markers of their efficiency; this could be coupled with circulating tumoral DNA monitoring, such as dynamic evaluation of primary or acquired resistance [119].

### 4.3. Recurrent Epithelial Ovarian Cancer Perspectives

Currently, FDA-approved PARPis for recurrent epithelial ovarian cancer maintenance are independent of biomarkers. This highlights several hypotheses: either HRD could be irrelevant within this context and/or PARPis could exert functions in an HRD-unrelated manner. Furthermore, it is also possible that current CDx do not appropriately assess HRD status. Thus, the nomogram is a useful tool for treatment choice. To develop higher nomogram accuracy, a more comprehensive version should integrate the type of *BRCA* alteration (if applicable) and other dynamic biomarkers. For instance, integrating evolutionary perspectives with circulating tumoral DNA for detecting reverse mutations and RAD51 foci for determining the actual HRD status could lead to a more relevant evaluation of HRD status and consequently to better patient management.

With respect to third-line treatments and beyond, a deeper analysis is needed of the correlation between the type of *BRCA1/2* alteration and PARPi sensitivity, as all patients treated with olaparib and rucaparib do not exhibit a response [62]. The development of a “clinicomolecular” nomogram could be useful for clinical practice. Strikingly, HRD does not appear to influence niraparib efficiency in the context of *BRCA^wt^*; consequently, more accurate biomarkers are urgently needed.

An evolutionary perspective should also be considered regarding HRD scores such as LOH and GIS. Indeed, these scores should change between primary sampling upon diagnosis and recurrence/metastasis, and in the case of HRD, evolve higher values. Therefore, studies should evaluate GIS dynamics over time, especially in patients already treated with platinum and PARPis and with a higher risk of acquired resistance. Furthermore, a refined GIS cutoff should be determined depending on the clinical context, rather than using ≥42, which is designed for naïve patients.

The OReO ENGOT-Ov-38 trial (NCT03106987) focused on olaparib maintenance retreatment versus placebo. It enrolled heavily pretreated patients separated into two cohorts by *BRCA* status; 112 patients were *BRCA** (74 and 38 received olaparib and placebo, respectively), while 108 patients were *BRCA^wt^* (72 and 36 received olaparib and placebo, respectively). Interestingly, this trial reported positive results irrespective of *BRCA* status. Indeed, the *BRCA** cohort exhibited a median PFS of 4.3 (olaparib) versus 2.8 months (placebo) (HR = 0.57; 95% CI: 0.37–0.87; *p* = 0.022), while *BRCA^wt^* showed a median PFS of 5.3 (olaparib) versus 2.8 months (placebo) (HR = 0.43; 95% CI: 0.26–0.71; *p* = 0.002). This seminal work paved the way for “PARPi rechallenge” [120] and raised new questions regarding the deciphering of predictive markers.

## 5. Synthesis and Concluding Remarks

HGSOC is the most frequent and aggressive form of ovarian cancer, thus representing an important challenge for translational researchers and clinicians. HRD, an altered pathway present in half of the cases, is predictive of sensitivity to PARPis, a novel class of molecules that have led to substantial improvements in prognosis. HRD is mainly caused by genetic alterations in *BRCA1/2* genes, although other causes (encompassed under the concept of “*BRCAness*”) should not be overlooked. Following seminal clinical trials that exhibited vast improvements in prognosis, three PARP inhibitors (olaparib, niraparib and rucaparib) received FDA/EMA approval. Currently validated CDx assays suffer from several technical and medical limitations that need to be investigated for more relevant integration within clinical applications. While used to determine frontline maintenance treatment, HRD evaluation and its relevance as a predictive biomarker still remains a matter of debate for clinicians. Several axes of research, including integrating tumor heterogeneity and PARP inhibitor resistance, are currently under development. Promising strategies, including functional assays that evaluate present HRD status, should be integrated into clinical studies in the near future. Beyond a simplistic dichotomy (i.e., proficient/deficient status), future studies assessing HRD should consider its causes and consequences from an evolutionary perspective.

## Figures and Tables

**Table 1 cancers-14-01098-t001:** Current FDA/EMA-approved PARPis in epithelial ovarian cancer according to the molecular context.

Molecular Status ^1^	Advanced Epithelial Ovarian Cancer in Complete/Partial Response to Platinum First-Line Maintenance Monotherapy ^2^	Recurrent Epithelial Ovarian Cancer in Complete/Partial Response to Platinum Second-Line Maintenance Monotherapy ^2^	Recurrent Epithelial Ovarian Cancer ≥Third-Line (3L) Monotherapy
*gBRCA**	**Olaparib** (FDA 2018/EMA 2019; SOLO-1)—BA-CDx		**Olaparib** after ≥3L—regardless of platinum sensitivity ^3^ (FDA 2014; STUDY-42)—BA-CDx
*tBRCA**	**Olaparib** (FDA 2018/EMA 2019; SOLO-1)—F1-CDx		**Rucaparib** after ≥2L—regardless of platinum sensitivity ^4^ (FDA 2016/EMA 2019; ARIEL-2/STUDY-10)—F1-CDx
HRD+	**Olaparib + bevacizumab** (FDA 2020/EMA 2020; PAOLA-1)—MC-CDx		**Niraparib** after ≥3L—potentially platinum sensitive ^5^ (FDA 2019; QUADRA)—MC-CDx
Biomarker agnostic	**Niraparib** (FDA 2020/EMA 2020; PRIMA)	**Niraparib** (FDA 2017/EMA 2017; NOVA) ^6^ **Olaparib** (FDA 2017/EMA 2018; SOLO-2/STUDY-19) **Rucaparib** (FDA 2018/EMA 2019; ARIEL-3)—F1-CDx ^7^	

Abbreviations: *gBRCA* = germline *BRCA*; HRD+ = homologous recombination deficiency positive; NA = not applicable; *tBRCA* = tumoral *BRCA*. Remarks: The dates within parentheses correspond to the years of FDA/EMA approvals, followed by the main study that led to these approvals. Underlined text represents FDA-approved CDx. All EMA approvals are for high-grade epithelial ovarian cancers only; furthermore, according to the EMA, niraparib second-line maintenance requires a platinum-sensitive status. Iterative challenges with PARPis have not been validated thus far. The term “epithelial ovarian cancer” includes primitive peritoneal and fallopian tube cancers. ^1^ By definition, an approval for HRD+ (corresponding to *tBRCA+* and/or genomic instability score positivity) includes *gBRCA** and *tBRCA**; similarly, an approval for *tBRCA** includes *gBRCA**; for clarity, the approval is provided only once, corresponding to the broadest situation. ^2^ All maintenance therapies were approved in the context of complete/partial response to platinum-based chemotherapy; furthermore, all PARPis except olaparib + bevacizumab were used as monotherapies. ^3^ This approval is restricted to ovarian cancers only. ^4^ Concerning rucaparib in the third-line setting, EMA approval diverges; it is restricted to the third line and for patients who are unable to tolerate further platinum-based chemotherapy. ^5^ In the context of niraparib, HRD is defined by (1) a *tBRCA* mutation or (2) GIS ≥ 42 in patients with cancer progression more than six months after response to the last platinum-based chemotherapy. ^6^ EMA approval of niraparib is restricted to epithelial ovarian cancers with platinum-sensitivity in this context. ^7^ Rucaparib approval in this context is not biomarker driven, but F1-CDx has FDA approval, as a positive HRD status is predictive of efficacy and indicates enhanced progression-free survival.

**Table 2 cancers-14-01098-t002:** Technical considerations associated with validated CDx for HRD assessment.

General Consideration	Distinct CDx Are Not Interchangeable
**Preanalytical considerations**	≈5–10% of specimens are inadequate Sample heterogeneity: Tumoral versus normal Intratumoral heterogeneity
**Analytical considerations**	Limit of detection *(BRCA1/2*): >20% tumor cellularity for SNVs >30–35% tumor cellularity for genomic scars SVs are poorly detected (apart from BA-CDx)
**Post-analytical considerations**	≈5–10% of results are inconclusive GIS positivity thresholds: FP (i.e., indicated HRD+ but actually HRD−) FN (i.e., indicated HRD− but actually HRD+) *BRCA* mutations (VUS versus pathogenic status)

Abbreviations are as follows: BA-CDx = BRACAnalysis CDx; CDx = companion diagnostic; GIS = genomic instability score; FN = false negative; FP = false positive; HRD (+/−) = homologous recombination deficiency (positive/negative); SNV = single nucleotide variant; SV = structural variant; VUS = variant of unknown significance.

**Table 3 cancers-14-01098-t003:** Medical considerations associated with validated CDx for HRD assessment.

General Consideration	Cost and Access to HRD Assays Currently Restricted to Private Companies
**Tissue heterogeneity of HGSOC**	Sample heterogeneity: Intratumoral Primitive tumor versus metastases (Epi)genomic context Cellular microenvironment effect
**Relevance as a biomarker** **(PARPi sensitivity)**	Patient selection and clinical context (e.g., platinum-sensitivity status) Timing of analysis Limited predictivity of GIS: Near-threshold scores Beyond frontline treatment Out of platinum-sensitive context
**Evolutionary perspective of HGSOC**	PARPi resistance: reverse mutations, HRD-unrelated mechanisms Genomic scars are irreversible Iterative analysis and PARPi rechallenge

Abbreviations are as follows: GIS = genomic instability score; HGSOC = high-grade serous ovarian cancer; HRD = homologous recombination deficiency; PARPi = poly(adenosine diphosphate-ribose) polymerase inhibitor.

**Table 4 cancers-14-01098-t004:** Emerging strategies for iterative HRD evaluation.

	Advanced Epithelial Ovarian Cancer —First-Line Maintenance	Recurrent Epithelial Ovarian Cancer
Shown to be associated with PARPi sensitivity	Molecular assays (e.g., HRDetect, *RAD51C* methylation, gene amplification, SVs, non-coding RNAs, transcriptomics, proteomics) Functional assays (e.g., RAD51 foci)	Reverse mutations (e.g., in *BRCA1/2*) Nomogram
Research and future strategies	Stepwise approach (i.e., integrate other biomarkers in inconclusive cases) Deeper refinement of HRD status (e.g., type of *tBRCA**, GIS thresholds, tumor heterogeneity) Comprehensive PARPi sensitivity score Integrating clinical and biopathological data Through-treatment dynamic markers	More accurate and comprehensive evaluation, such as: Type of alteration that initially caused HRD Context-dependent HRD positivity thresholds GIS dynamics Mutations in other HRR genes Functional assays (e.g., RAD51 foci)

Abbreviations: GIS = genomic instability score; HRD = homologous recombination deficiency; HRR = homologous recombination-related; PARPi = PARP inhibitor; SV = structural variant; *tBRCA* = tumoral *BRCA*.

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
