# Peer review of "Toward More Comprehensive Homologous Recombination Deficiency Assays in Ovarian Cancer Part 2: Medical Perspectives"

_cancers, 2022, doi:10.3390/cancers14041098_

Round 1

Reviewer 1 Report

The authors provide a well written comprehensive review of HRD deficiency selection for PARPui and the three approved tests. There are minor issues below

Specific points

  1. It is somewhat unusual when talking about HRD deficiency and ‘certain molecular changes’ to not even mention BRCA1 and BRCA2 in the abstract.
  2. ‘EOCs have been divided into type I and II cancers: type I includes endometriosis-related tumors (i.e., endometrioid and clear cell carcinomas), mucinous and low-grade serous carcinomas, while type II includes HGSOCs, undifferentiated carcinomas and carcinosarcomas’ -Neither undifferentiated carcinomas and carcinosarcomas were mentioned in the sentence above regarding proportions of EOC please add to the previous sentence
  3. ‘In terms of etiology, the main cause of HRD is a mutation in the BRCA1 or BRCA2 (BRCA1/2) genes’ -Not strictly true. The main cause is bi-allelic inactivation of BRCA1 or BRCA2 by either germline plus somatic loss or loss of both functional alleles somatically. Although it is possible haploinsufficiency may play a role HRD is caused by no active BRCA1 or BRCA2 protein
  4. ‘Briefly, BRCAness can be caused by biallelic mutations in some homologous recombination-related (HRR) genes,EMSY amplification or epigenetic alterations [19–25].’- Please list the genes
  5. ‘Part 2 aim(s) to bridge the gap between technical and clinical perspectives. Thus, we will review: 1
  6. ‘Clinically, HRD tumors tend to be more sensitive to platinum-based regimens, and specifically to PARPis, through synthetic lethality [13,35].’ -In case many readers do not read part 1 first please provide a short definition of what synthetic lethality is.
  7. ‘To date, 3 CDx assays are currently FDA-approved for EOC cases [37].’ Please list them
  8. ‘BRACAnalysis® CDx (BA-CDx), developed and marketed by MyriadGenetics (MG),relies on gBRCA assessment through sequencing genomic DNA obtained from whole blood samples collected in EDTA [38].’ -Many clinical and commercial labs carry out gBRCA analysis. Are these not covered by FDA approval?
  9. ‘Notably, a positive HRD result is indicated if the tumor is “LOH high” and/or exhibits a tBRCA*.’ -I have seen HRD reports that show loss of one copy of BRCA and are HRD-. Could this tBRCA* be just random LOH and therefore not indicate a targeted aberration of BRCA and therefore no sensitivity to PARPi?
  10. ‘In the entire sample, niraparib increased the median PFS (13.8 months with niraparib versus 8.2 months with placebo; HR= 0.62; 95% CI: 0.50-0.76; p<0.001). This latter point led to FDA/EMA approvals of 1Lm niraparib irrespective of biomarker status.’ -Yes but what is the difference in HRD- cases? Surely this should have been assessed before an FDA decision that HRD testing was not required?
  11. ‘Overall, PARPis as 1Lm treatment has become the standard according to American and European clinical guidelines: olaparib (in-is) recommended in the context of tBRCA* and olaparib plus bevacizumab is recommended in HRD+ tumors.’ Replace ‘in’ with ‘is’
  12. ‘Given the superiority of tBRCA* as a predictor…’ -Over BRCA wild type HRD+ or over gBRCA?? Please clarify
  13. ‘The further a patient is from frontline treatment, the higher the risk of discordance between GIS and actual status, as therapies mainly rely on platinum, which will artificially select clones that are resistant to DSBs’ -Please clarify. Platinum selectively treats clones that have DSBs so it is not that platinum ‘selects’ the non DSB clones but platinum treatment means that these clones are spared

Author Response

The authors provide a well written comprehensive review of HRD deficiency selection for PARPi and the three approved tests. There are minor issues below

 The authors sincerely thank the reviewer for the interesting remarks; we will assess it point by point: 

Specific points

  1. It is somewhat unusual when talking about HRD deficiency and ‘certain molecular changes’ to not even mention BRCA1 and BRCA2 in the abstract.

Thanks for this remark, we added this precision 

  1. ‘EOCs have been divided into type I and II cancers: type I includes endometriosis-related tumors (i.e., endometrioid and clear cell carcinomas), mucinous and low-grade serous carcinomas, while type II includes HGSOCs, undifferentiated carcinomas and carcinosarcomas’ -Neither undifferentiated carcinomas and carcinosarcomas were mentioned in the sentence above regarding proportions of EOC please add to the previous sentence

Thanks for this remark, we added this precision  “undifferentiated carcinomas and carcinosarcomas (these two last classes represent less than 1% of OC and are not EOC per se).”

  1. ‘In terms of etiology, the main cause of HRD is a mutation in the BRCA1 or BRCA2 (BRCA1/2) genes’ -Not strictly true. The main cause is bi-allelic inactivation of BRCA1 or BRCA2 by either germline plus somatic loss or loss of both functional alleles somatically. Although it is possible haploinsufficiency may play a role HRD is caused by no active BRCA1 or BRCA2 protein

Thanks for this interesting point, we actually added this sentence in order to provide the readers with a clearer view : In terms of etiology, the main cause of HRD is bi-allelic inactivation of BRCA1 or BRCA2 (BRCA1/2) genes by either germline (gBRCA*) plus somatic (sBRCA*) loss or both functional alleles somatically. Short mutations (single nucleotide or short insertions/deletions) in BRCA1/2 within the tumor (tBRCA*), which can reveal either gBRCA* or sBRCA* are estimated to be present in approximately 20-25% of cases

  1. ‘Briefly, BRCAness can be caused by biallelic mutations in some homologous recombination-related (HRR) genes,EMSY amplification or epigenetic alterations [19–25].’- Please list the genes

Thanks for this interesting point, we actually added this sentence “Apart from BRCA1/2, the following list of HRR genes is currently considered as the best characterized to date: ATM, BARD1BRIP, CDK1, PALBCHEK1, CHEK2, FANCL, PPP2R2ARAD51B, RAD51C, RAD51D and RAD54L”

  1. ‘Part 2 aim(s) to bridge the gap between technical and clinical perspectives. Thus, we will review: 1

Sorry but we do not understand this remark

  1. ‘Clinically, HRD tumors tend to be more sensitive to platinum-based regimens, and specifically to PARPis, through synthetic lethality [13,35].’ -In case many readers do not read part 1 first please provide a short definition of what synthetic lethality is.

Thanks for this remark, we added this sentence

Briefly, SL relies on the fact that cancer cells harbor gene defects which are not lethal per se, but which turn lethal when combining with a defect in another gene. Using of PARPi leads to SSB accumulation which progress to DSB. In the context of HRD, cells will accumulate DSB, ultimately leading to apoptosis.

  1. ‘To date, 3 CDx assays are currently FDA-approved for EOC cases [37].’ Please list them

Thanks for this remark, we listed them

  1. ‘BRACAnalysis® CDx (BA-CDx), developed and marketed by MyriadGenetics (MG),relies on gBRCA assessment through sequencing genomic DNA obtained from whole blood samples collected in EDTA [38].’ -Many clinical and commercial labs carry out gBRCA analysis. Are these not covered by FDA approval?

 Actually, BA-CDx is the only CDx specifically assessing gBRCA that is FDA-approved and which is used in the clinical trials developed underneath.

  1. ‘Notably, a positive HRD result is indicated if the tumor is “LOH high” and/or exhibits a tBRCA*.’ -I have seen HRD reports that show loss of one copy of BRCA and are HRD-. Could this tBRCA* be just random LOH and therefore not indicate a targeted aberration of BRCA and therefore no sensitivity to PARPi?

 Actually, mechanistic consideration regarding discrepancy between BRCA status and LOH or GIS is a matter of huge debate. Indeed, BRCA haploinsufficiency is still a matter of consideration regarding its potency to create HRD. Furthermore, discrepancy can be the consequence of gneomic context. Indeed, for instance, GIS has been developed to be able to detect through the “magic 42 cutoff” 95% of BRCA mutated cases. Beyond this, HRD is not the only carcinogenetic process that can lead to  LOH.  

  1. ‘In the entire sample, niraparib increased the median PFS (13.8 months with niraparib versus 8.2 months with placebo; HR= 0.62; 95% CI: 0.50-0.76; p<0.001). This latter point led to FDA/EMA approvals of 1Lm niraparib irrespective of biomarker status.’ -Yes but what is the difference in HRD- cases? Surely this should have been assessed before an FDA decision that HRD testing was not required?

Thanks for the really interesting point, we added this sentence : In patients with HRD negative tumors, the PFS was still slightly higher with 8.1 months (versus 5.4 months with placebo; HR= 0.68).

  1. ‘Overall, PARPis as 1Lm treatment has become the standard according to American and European clinical guidelines: olaparib (in-is) recommended in the context of tBRCA* and olaparib plus bevacizumab is recommended in HRD+ tumors.’ Replace ‘in’ with ‘is’

Thanks for the remark, it has been changed

  1. ‘Given the superiority of tBRCA* as a predictor…’ -Over BRCA wild type HRD+ or over gBRCA?? Please clarify

Thanks for the remark, we changed to “Given the superiority of tBRCA* over HRD as a predictor of longer PFS”

  1. ‘The further a patient is from frontline treatment, the higher the risk of discordance between GIS and actual status, as therapies mainly rely on platinum, which will artificially select clones that are resistant to DSBs’ -Please clarify. Platinum selectively treats clones that have DSBs so it is not that platinum ‘selects’ the non DSB clones but platinum treatment means that these clones are spared

Thanks for this very interesting remark. Actually, we changed the sentence to “…as therapies mainly rely on platinum. Indeed, as platinum salts generate DSBs, they will treat clones with HRD but HR proficient clones will be spared.”

Reviewer 2 Report

This review summarizes the clinical trials with PARP inhibitors in ovarian cancer. It is a worthwhile effort to discuss the results that have been obtained in rather diverse settings and with different PARP inhibitors. Although similar reviews appeared recently, this one could add to clarify some issues.

The main point that remains after reading the manuscript is that the term HRD is a bit confusing. The authors begin to question the meaning of this term towards the end of the review, but it would be really helpful if they could already discuss this to some extent in the beginning of the review. The term is now mainly used to describe the outcome of some commercial tests that measure chromosomal rearrangements or loss of chromosomal regions (LOH). However, this is not directly measuring homologous recombination, but rather observations that correlate with BRCA gene mutations, and by extension with HRD. It would be helpful to clarify this already at the beginning, so readers have a clear idea of what the term means in this context. I would be helpful to stress that ideally, one would like to measure HR capacity directly, but that this is not practically possible.

Minor points:

Line 93-94: genomic scars are not only large scale rearrangements, but also small deletions or insertions with specific characteristics, such as microhomology at the junction.

Line 104: Why is HRD status essential for ‘theranostics’? I assume that the authors mean that it can be predictive for therapy response, but I don’t think the term theranostics is correct for this.

Line 166: the authors refer to reference 46, which is one of the two accompanying papers that were back to back in Nature. They should also refer to the other one.

Line 171-172: it would be helpful to mention which three PARPi have been approved already here, probably with the companies that market them.

Line 215: delete ‘several possibilities,’.

Line 429: ‘in by an’ should be ‘in an’.

Line 443-444: ‘that could modulate the functional effects of HRD’ (of should be added).

Line 474-475: it is not so clear that ambiguity in BRCA mutation stems form HRD-unrelated functions of BRCA. Other options would be that it reduces HR, but that it depends on other factors whether that is so low that it really affects the overall HR capacity or that the mutation in combination with the exact expression level is important. I would just delete this sentence. The authors again mention these HRD unrelated pathways in line 529. However, without a discussion what that means, it is not really understandable.

Line 609: ‘ESMY amplification’ should be ‘EMSY amplification’.

Line 634-635: ‘HRD is useless within this context, and/or 634 PARPis exert functions in an HRD-unrelated manner’. This is not necessarily correct: it is also possible that the HRD assays do not really identify HRD in the tumors and we need better assays. The authors come back to this a few sentences later, but they should not suggest here that there are only two possibilities.

Author Response

The authors sincerely thank the reviewer for the interesting remarks; we will assess it point by point:  

This review summarizes the clinical trials with PARP inhibitors in ovarian cancer. It is a worthwhile effort to discuss the results that have been obtained in rather diverse settings and with different PARP inhibitors. Although similar reviews appeared recently, this one could add to clarify some issues.

 The main point that remains after reading the manuscript is that the term HRD is a bit confusing. The authors begin to question the meaning of this term towards the end of the review, but it would be really helpful if they could already discuss this to some extent in the beginning of the review. The term is now mainly used to describe the outcome of some commercial tests that measure chromosomal rearrangements or loss of chromosomal regions (LOH). However, this is not directly measuring homologous recombination, but rather observations that correlate with BRCA gene mutations, and by extension with HRD. It would be helpful to clarify this already at the beginning, so readers have a clear idea of what the term means in this context. I would be helpful to stress that ideally, one would like to measure HR capacity directly, but that this is not practically possible.

Regarding this first point, this review is actually a kindred review with a bench to bedside approach and the whole and extensive description of HRD mechanisms is developed in the first part of the review. We added the sentence in the introduction : “To meet this demand, several companion diagnostic (CDx) assays have been developed and clinically validated, leading to substantial improvements in the management of HGSOC. However, their use remains controversial, notably because they do not assess directly HRD status but their causes and/or their consequences …

Minor points:

Line 93-94: genomic scars are not only large scale rearrangements, but also small deletions or insertions with specific characteristics, such as microhomology at the junction.

Thanks for the interesting remark, actually genomic scars encompass any HRD-related DNA alteratyion. Meanwhile, in the common usage it refers to lrge scale alterations as CDx assays such as  MyriadGenetics one only assess these.

We added the sentence : Noteworthy, although “genomic scars” refers to any HRD-related DNA mutation (i.e. encompassing both microlesions and macrolesions), we will restraint this term to large-scale rearrangements as it is used in the literature.

Line 104: Why is HRD status essential for ‘theranostics’? I assume that the authors mean that it can be predictive for therapy response, but I don’t think the term theranostics is correct for this.

Thanks for the interesting remark, actually the term Theranostics could appear a bit confusing so we changed it for a more neutral one “ is essential for both prognosis and therapeutic choice”.

Line 166: the authors refer to reference 46, which is one of the two accompanying papers that were back to back in Nature. They should also refer to the other one.

Thanks for the interesting remark, we added this seminal work as a reference.

Line 171-172: it would be helpful to mention which three PARPi have been approved already here, probably with the companies that market them.

Line 215: delete ‘several possibilities,’.

Thanks for the remark, we deleted the duplicate

Line 429: ‘in by an’ should be ‘in an’.

Thanks for the remark, we did the correction

Line 443-444: ‘that could modulate the functional effects of HRD’ (of should be added).

Thanks for the remark, we added the word which was lacking

Line 474-475: it is not so clear that ambiguity in BRCA mutation stems form HRD-unrelated functions of BRCA. Other options would be that it reduces HR, but that it depends on other factors whether that is so low that it really affects the overall HR capacity or that the mutation in combination with the exact expression level is important. I would just delete this sentence. The authors again mention these HRD unrelated pathways in line 529. However, without a discussion what that means, it is not really understandable.

Thanks for the very interesting remark, although HRD-unrelated effect of PARPi is developed in the first part of the review, their impact within clinics still remain controversial and indeed would need deeper description. As such, we removed these sentences.

Line 609: ‘ESMY amplification’ should be ‘EMSY amplification’.

Thanks for the remark, we did the correction

Line 634-635: ‘HRD is useless within this context, and/or 634 PARPis exert functions in an HRD-unrelated manner’. This is not necessarily correct: it is also possible that the HRD assays do not really identify HRD in the tumors and we need better assays. The authors come back to this a few sentences later, but they should not suggest here that there are only two possibilities.

Thanks for the interesting remark, we modulated the sentence to “ This highlights several hypotheses: either HRD could be irrelevant within this context and/or PARPis could exert functions in an HRD-unrelated manner. Furthermore, it is also possible that current CDx do not appropriately assess HRD status.

Reviewer 3 Report

This manuscript summarizes the results of a series of studies that investigated new therapeutic and diagnostic approaches for malignant tumors of the ovary, with particular regard to carcinomas. Although it may certainly be of use to oncologists, the manuscript is difficult to read and comprehend in full for other physicians and, mostly, for non-medical researchers. Perhaps, if the description of the diagnostic and therapeutic strategies were accompanied by an overview of the cellular and molecular events underlying the onset and clinical progression of ovarian carcinomas, this article would be "digestible" by a wider audience. A strong reduction in the use of abbreviations (currently exaggerated) would also help the comprehensibility of the text, making it more appealing.

Author Response

This manuscript summarizes the results of a series of studies that investigated new therapeutic and diagnostic approaches for malignant tumors of the ovary, with particular regard to carcinomas.

The authors sincerely thank the reviewer for this relevant comment. Actually, regarding the perspective proposed we would like to bring some concerns.

Although it may certainly be of use to oncologists, the manuscript is difficult to read and comprehend in full for other physicians and, mostly, for non-medical researchers.

Although it may appear difficult to comprehend for non oncologists, we guess that Cancers review is a high impact journal with translational approach, with the aim to provide oncologists, medical and non-medical researchers with up to date and comprehensive reviews. We think that our review (which is the part 2 of a kindred review) could help medical oncologists with a clear view of the current state of the art regarding HRD evaluation and PARPi prescription, as many existing reviews specifically assessed a fraction of this topic (i.e. HRD companion diagnostic assays description and performances, description of clinical trials assessing PARPi’s, PARPi/CDx assays prescription strategies). Furthermore, the two other reviewers did not raise concern regarding the potential difficulty for a wider audience.

Perhaps, if the description of the diagnostic and therapeutic strategies were accompanied by an overview of the cellular and molecular events underlying the onset and clinical progression of ovarian carcinomas, this article would be "digestible" by a wider audience. A strong reduction in the use of abbreviations (currently exaggerated) would also help the comprehensibility of the text, making it more appealing. 

HRD evaluation and PARPi prescription are hot trends in medical oncology community, with the need of comprehensive approach (as it has been recently highlighted by European experts consensus recommendations in late 2021) and an up to date view for 2022.

As explained  above, this is a kindred review with the first part specifically assessing molecular and cellular events related to HRD, its causes and consequences. As such, a lecture of both reviews appears necessary to get the holistic approach from molecule to patients. We did the choice of a kindred review in order to provide the readers with more technical considerations (part 1) and with a more medical ones (part 2).  As such, for researchers with a more “fundamental” approach, the part 1 will bring higher interest, while it will be a relevant premise for non oncologists. Notably, part 1 is enriched with figures for a more convenient approach and we decided to not reproduce them in this part 2.

Regarding the abbreviations, we do understand that they could appear exaggerated, but many of them are the ones currently and frequently used by medical community, and we did not want to lengthen the sentences and the review.

Furthermore, many reviews already provided researchers with comprehensive reviews specifically assessing molecular defects associated with ovarian cancers, and this perspective is out of scope of our review. Simply developing what is already found elsewhere appears less innovative and we decided to bring a “bench-to-bedside” perspective although we do understand that this could appear quite difficult for the readers that are not specifically working in this field.

For a more convenient perspective, the kindred review will be accompanied with a graphical abstract showing the main events related to HRD (with specific causes/consequences dichotomy).

Round 2

Reviewer 3 Report

Dear Authors,

Although I have been working in the field of oncology for many years, I found reading your manuscript very difficult and, sorry, not very engaging. Without a doubt this may be due, at least in part, to the fact that I was not shown the first part, which certainly served as a useful introduction and completion to this. Nevertheless, as first I must point out that your comments, whether critical or proactive, to the previously published data reviewed herein are reduced to a minimum. Secondly, I reiterate that the reading of your manuscript is severely hampered by the extensive use of abbreviations: if it is true that some of them are “currently and frequently used by medical community”, many others are not. Believe me, taken together, all these abbreviations make the manuscript appear to be written in a language other than English. I'm really sorry to have to write these things. My intention was to give you helpful suggestions to make your manuscript accessible to a wider audience. What else to say? If the other two Reviewers have given a favorable opinion, the Editors will publish your work in its present form.

Author Response

The authors kindly thank the reviewer for supporting comments. As such, we will revise the review and remove many abreviations used in order to make the manuscript accessible to a wider audience.

Sincerely yours